# Plasma Metabolites Associate with All-Cause Mortality in Individuals with Type 2 Diabetes

**DOI:** 10.3390/metabo10080315

**Published:** 2020-07-31

**Authors:** Filip Ottosson, Einar Smith, Céline Fernandez, Olle Melander

**Affiliations:** 1Department of Clinical Sciences, Lund University, 214 28 Malmö, Sweden; einar.smith@med.lu.se (E.S.); celine.fernandez@med.lu.se (C.F.); olle.melander@med.lu.se (O.M.); 2Department of Internal Medicine, Skåne University Hospital, 214 28 Malmö, Sweden

**Keywords:** metabolomics, diabetes, mortality, cardiovascular disease, N2,N2-dimethylguanosine and dimethylguanidino valerate

## Abstract

Alterations in the human metabolome occur years before clinical manifestation of type 2 diabetes (T2DM). By contrast, there is little knowledge of how metabolite alterations in individuals with diabetes relate to risk of diabetes complications and premature mortality. Metabolite profiling was performed using liquid chromatography-mass spectrometry in 743 participants with T2DM from the population-based prospective cohorts The Malmö Diet and Cancer-Cardiovascular Cohort (MDC-CC) and The Malmö Preventive Project (MPP). During follow-up, a total of 175 new-onset cases of cardiovascular disease (CVD) and 298 deaths occurred. Cox regressions were used to relate baseline levels of plasma metabolites to incident CVD and all-cause mortality. A total of 11 metabolites were significantly (false discovery rate (fdr) <0.05) associated with all-cause mortality. Acisoga, acylcarnitine C10:3, dimethylguanidino valerate, homocitrulline, N2,N2-dimethylguanosine, 1-methyladenosine and urobilin were associated with an increased risk, while hippurate, lysine, threonine and tryptophan were associated with a decreased risk. Ten out of 11 metabolites remained significantly associated after adjustments for cardiometabolic risk factors. The associations between metabolite levels and incident CVD were not as strong as for all-cause mortality, although 11 metabolites were nominally significant (*p* < 0.05). Further examination of the mortality-related metabolites may shed more light on the pathophysiology linking diabetes to premature mortality.

## 1. Introduction

The global epidemic of type 2 diabetes (T2DM) and its complications causes an enormous burden on society. In 2017 the worldwide prevalence of T2DM was estimated at 451 million and these numbers are projected to increase to almost 700 million by 2045 [1]. Individuals with T2DM develop cardiovascular disease (CVD) on average 14.6 years earlier [2] and twice as often [3] compared to individuals without T2DM. Moreover, diabetes is a major contributor to mortality, as it causes 5 million deaths yearly [4]. With this in mind, T2DM and its complications will remain a major contributor to morbidity and mortality in the coming decades. Despite these alarming numbers, the pathogenesis of diabetes complications, other than poor control of blood glucose, remain largely unknown [5]. 

Changes in the metabolome occur years before clinical manifestation of T2DM and cardiovascular disease (CVD), as several studies show that alterations of metabolites such as, branched-chain amino acids (BCAA) [6], aromatic amino acids (AAA) [6], glutamate [7,8] and dimethylguanidino valerate (DMGV) [9,10], associate with incidence of cardiometabolic disease (CMD). Moreover, non-targeted metabolomic studies have suggested alterations in a broad range of metabolic pathways in the onset of type T2DM [11,12]. Similarly, metabolite alterations in individuals with T2DM could help describing the pathogenesis of T2DM complications, but to our knowledge there is only one large prospective study focusing on describing how metabolite alterations in individuals with T2DM are related to future cardiovascular complications and mortality [13]. In that prospective cohort study by Welsh et al. investigating plasma levels of eight amino acids in individuals with T2DM, it was shown that two BCAA, leucine and valine, despite repeatedly being associated with increased risk of T2DM, were inversely associated with all-cause mortality [13]. Moreover, another known predictor of T2DM [6] and cardiovascular disease [14], tyrosine, was inversely associated with future microvascular complications and unrelated to future macrovascular complications [13]. Although the study by Welsh et al. highlights the potential of using metabolomics to find predictive biomarkers for diabetes complications, it suffers from the drawback of only measuring eight amino acids. 

In the present study, we have used metabolomics to measure 112 metabolites in more than 700 individuals with T2DM, from two different Swedish population-based prospective cohorts, The Malmö Diet and Cancer-Cardiovascular Cohort (MDC-CC) and the Malmö Preventive Project (MPP). Fasting plasma levels of metabolites measured at baseline are related to the future risk of all-cause mortality and cardiovascular disease.

## 2. Results

The baseline characteristics of the investigated cohorts MDC-CC and MPP can be found in Table 1. The participants in MPP (70.5 years) were older than in MDC-CC (59.5 years). A large majority of the participants in MPP were male (73.4%), as compared to MDC-CC were there only was a slightly higher proportion of male participants (55.1%). There was also a higher baseline prevalence of CVD in MPP (14.7%) compared to MDC-CC (4.5%), whereas there were only small differences in other CMD risk factors. 

The associations between plasma metabolite levels and all-cause mortality was investigated using Cox regression models. In meta-analysis of MDC-CC and MPP, 11 metabolites were significantly (false discovery rate (fdr) <0.05) associated with all-cause mortality (Figure 1) (Appendix A
Appendix A). Acisoga, acylcarnitine C10:3, DMGV, homocitrulline, N2,N2-dimethylguanosine, 1-methyladenosine and urobilin were associated with an increased risk for all-cause mortality, while hippurate, lysine, threonine and tryptophan were associated with a decreased risk. N2,N2-dimethylguanosine was the metabolite most strongly associated (hazard ratio (HR) = 1.45, *p* = 1.93 × 10^−5^, confidence interval (CI) = 1.22–1.72). 

Furthermore, the association between baseline levels of plasma metabolites and incident CVD was investigated in both MPP and MDC-CC using Cox regression models. In meta-analyses of MDC-CC and MPP, there were no metabolites with significant associations with incident CVD when using a fdr cut-off at 5%. Using a nominal *p*-value threshold (*p* < 0.05), there were 11 significantly associated metabolites. Glycerophosphocholine (HR = 0.77, CI = 0.66–0.90, *p* = 9.1 × 10^-4^) was the metabolite most strongly associated with a lower risk of CVD and the two BCAA valine (HR = 1.23, C.I = 1.05–1.44, *p* = 0.01) and isoleucine (HR = 1.20, C.I = 1.02–1.41, *p* = 0.02) were associated with an increased risk (Figure 2) (Appendix A
Appendix A). 

All metabolites that were significantly (fdr <5%) associated with mortality were investigated further by examining their correlation with traditional CMD risk factors in both MDC-CC (Figure 3) and MPP (Figure 4). The correlations between mortality-related metabolites and fasting glucose levels were weak. Lysine was the only metabolite which levels correlated with fasting glucose in both MDC-CC and MPP. Interestingly, lysine was positively correlated with fasting glucose levels despite being associated with decreased risk of all-cause mortality. N2,N2-dimethylguanosine was positively correlated with body mass index (BMI) and inversely correlated with high-density lipoprotein (HDL) cholesterol in both MDC-CC and MPP. Particularly strong correlations were seen between levels of DMGV and several CMD risk factors, including positive correlations with BMI and triglycerides and inverse correlations with HDL cholesterol. Moreover, in MDC-CC, glycosylated hemoglobin (Hba1c) and fasting insulin levels were measured and insulin resistance was estimated using homeostatic model assessment (HOMA-IR). DMGV levels were strongly correlated with HOMA-IR (rho = 0.40, *p* < 0.001) but were uncorrelated with fasting Hba1c. Lysine (rho = 0.18, *p* < 0.001) and C10:3-carnitine (rho = 0.14, *p*=9.0 × 10^−3^) showed significant positive correlations with Hba1c, while threonine (rho = −0.17, *p* < 0.001) was inversely correlated. All correlations with Hba1c and HOMA-IR are found in Appendix A
Appendix A. 

Since several metabolites were correlated with CMD risk factors, Cox regression models on all-cause mortality were further adjusted for fasting glucose, BMI, HDL cholesterol, LDL cholesterol, triglycerides, systolic blood pressure, anti-hypertensive treatment and smoking status. All metabolites except acisoga (HR = 1.11, CI = 0.98–1.25, *p* = 0.11) remained significantly (*p* < 0.05) associated after full adjustments for CMD risk factors (Table 2). 

## 3. Discussion

The key findings of the present study is that plasma levels of 10 metabolites, including N2,N2-dimethylguanoisine and DMGV, associate with all-cause mortality in individuals with diabetes, independently of CMD risk factors. These metabolites highlight novel potential pathophysiological links between diabetes and premature mortality.

Metabolomics has been used to find alterations in the metabolism that are associated with future onset of diabetes [6,7]. These alterations that are present years before clinical manifestation might help in assessing an individual’s risk of developing T2DM but are not necessarily related to health outcomes in individuals with overt diabetes. We recently reported that N2,N2-dimethylguanosine [11] was associated with an increased risk of future T2DM and DMGV with an increased risk of both T2DM and coronary artery disease [10] in individuals free from baseline cardiometabolic disease in the MDC-CC and MPP, independent of traditional risk factors. Moreover, N2,N2-dimethylguanosine has also been associated with an increased risk of all-cause mortality in postmenopausal women [15]. We now extend these findings by showing that levels of N2,N2-dimethylguanosine and DMGV also are associated with increased risk of all-cause mortality in individuals with diabetes. Both DMGV [10,16] and N2,N2-dimethylguanosine [11,17] have previously been shown to correlate strongly with CMD risk factors, such as obesity and dyslipidemia in healthy individuals. Furthermore, we showed that DMGV is very strongly correlated with HOMA-IR in individuals with diabetes. These findings indicate that DMGV and N2,N2-dimethylguanosine could be considered as markers for metabolic health, not only in the diabetes-free population, but that these correlations persist in diabetes. Despite these strong correlations with markers of metabolic health, their associations with all-cause mortality were independent of such risk factors. 

Levels of lysine [18], threonine [18] and tryptophan [15] have previously been associated with longevity and are here shown to be inversely associated with all-cause mortality. Interestingly, lysine has also been associated with hyperglycemia, obesity and future risk of T2DM [7]. We show here that among individuals with diabetes, lysine levels are positively correlated with fasting glucose levels and Hba1c, but simultaneously associated with a decreased risk of all-cause mortality. 

Although the associations between plasma metabolite levels and incident CVD were weaker than what could be observed for all-cause mortality, 11 metabolites showed associations that were nominally significant. Interestingly, two BCAA that previously have been shown to associate with incident T2DM [6] and CVD [19,20], valine and isoleucine, were associated with an increased risk of CVD in individuals with T2DM. This indicates that the BCCA not only can predict future diabetes, but also can be indicative of CVD risk in individuals with overt diabetes. Three metabolites that previously have been shown to be markers of fruit intake, N-methylproline [21], proline betaine and hippurate [22] were associated with a lower risk of incident CVD. Moreover, hippurate was associated with lower risk of all-cause mortality, even when adjusting for traditional risk factors. N-methylproline was recently shown to associate with healthy eating, to be decreased after following the Dietary Approaches to Stop Hypertension (DASH) diet [23] and to associate with decreased risk of CAD and all-cause mortality in individuals free from CMD [24]. In concert with these previous findings, our study indicates that healthy eating, particularly intake of citrus fruit, is beneficial in individuals with diabetes. We speculate that these effects can be partially driven by health-promoting metabolites, such as hippurate, N-methylproline and proline betaine. 

The present study has several limitations. The relatively small sample size of the present study encourages further studies to replicate the findings. Furthermore, although investigating over 100 annotated plasma metabolites, it is possible that clinically important metabolites could be found by increasing the metabolite coverage, most importantly by measuring lipids, which are not measured in the present study. Lastly, given the important role of diet in type 2 diabetes, including dietary assessment in future studies could help elucidate the relationship between plasma metabolites, dietary intake, diabetes complications and survival. 

## 4. Materials and Methods

### 4.1. Study Samples

Plasma metabolite levels were measured in a total of 743 participants from two different Swedish population-based prospective cohorts, MPP and MDC-CC.

MPP is a population based prospective cohort of 33,346 individuals, enrolled between 1974 and 1992. Between 2002 and 2006, 18,240 individuals (65–80 years old) were re-examined for cardiometabolic risk factors and overnight-fasting EDTA plasma was collected and stored at −80 °C for later analyses. A sub-cohort was created from a random sample of 5386 individuals from the re-examination that was selected for more detailed phenotyping. Metabolomic analysis was performed on the blood plasma samples from all 369 individuals in this sub-cohort who were either diagnosed with diabetes prior to or at the baseline examination. Among participants with diabetes at baseline, 54 participants had a history of cardiovascular disease (CVD) and 57 developed CVD within an average follow-up time of 5.4 years. A total of 118 deaths occurred within the average follow-up time of 7.7 years. 

MDC-CC is a population-based cohort, designed to study the epidemiology of carotid artery disease with participants being enrolled between 1991 and 1996. Among the 5405 participants who fasted, citrate plasma was available for 3799 participants. A total of 374 participants were either diagnosed with diabetes prior to or during the baseline examination. Among these participants, 17 had a history of prior CVD and 118 developed CVD within an average follow-up time of 17.5 years. During an average follow-up time of 18.3 years, a total of 180 deaths occurred. 

The ethics committee of Lund University approved the study protocols for MPP and MDC (DNR 2009/633), and all participants provided written informed consent. 

### 4.2. Endpoint Definitions and Biochemical Measurements

CVD was defined as fatal or non-fatal stroke, fatal or non-fatal myocardial infarction or death due to ischemic heart disease. The study subjects were followed for incident CVD through record-linkage using the Swedish personal identification number with the previously validated Swedish Hospital Discharge Register, the Swedish Cause of Death Register and the Swedish Coronary Angiography and Angioplasty Registry (SCAAR) [25]. T2DM was defined as a fasting plasma glucose of >7.0 mmol/L or a history of physician diagnosis of T2DM or being on antidiabetic medication or having been registered in local or national Swedish diabetes registries [26]. International Classification of Diseases (ICD) codes and details about biochemical measurements are found in Appendix A. 

### 4.3. Analytical Procedure

Profiling of plasma metabolites was performed using a UPLC-QTOF-MS System (Agilent Technologies 1290 LC, 6550 MS, Santa Clara, CA, USA) and has previously been described in detail [27]. Briefly, plasma samples stored at −80 °C were thawed and extracted by addition of six volumes of extraction solution. The extraction solution consisted of 80:20 methanol/water containing isotope labeled internal standards for 33 metabolites (Appendix A). Extracted samples were separated on an Acquity UPLC BEH Amide column (1.7 μm, 2.1 mm × 100mm; Waters Corporation, Milford, MA, USA) before being analyzed in positive ion mode. Samples were analyzed in batches of 180 samples, where quality control samples were run in the beginning of each batch and every eight analytical sample, in order to condition the column and to capture analytical drift, respectively. Metabolites were annotated using synthetic standards or by matching MS/MS fragmentation with The Human Metabolome Database (HMDB) [28] and METLIN [29] or by matching fragment ions to putative molecule fragments. In total, 112 Identified and putatively annotated metabolites were measured. A list of reported metabolites can be found in Appendix A, including corresponding mass, chromatographic retention time, molecular formula, HMDB identifier, annotation confidence [30] and analytical coefficient of variation.

### 4.4. Data Processing

Metabolite peak areas were integrated using Agilent Profinder B.06.00 (Agilent Technologies, Santa Clara, CA, USA). The allowed ion adducts included (M+H)^+^ and (M+NH_4_)^+^. Quality control samples were injected every eight analytical samples, in order to ensure high analytical repeatability. A total of 33 metabolites were normalized to the heavy-isotope labeled internal standards and the remaining metabolites were normalized using measurements in the quality control samples using low-order nonlinear locally estimated smoothing functions (supplementary material 2) [31]. Information about which normalization method that was used for each measured metabolite is presented in Appendix A. 

### 4.5. Statistical Analysis

In MDC-CC and MPP, the association between baseline levels of circulating metabolites and incident CVD and all-cause mortality were analyzed using Cox proportional hazards models. In regression analyses of incident CVD, all participants with a history of prevalent CVD were excluded. In model 1, regression models were adjusted for age and sex. Inverse-variance weighted meta-analyses (fixed intercepts) of Cox proportional hazard estimates from MDC-CC and MPP were performed for all metabolites using model 1. All metabolites that were significant (false-discovery rate < 0.05) in meta-analysis were subsequently analyzed using model 2, which was further adjusted for systolic blood pressure, smoking status, anti-hypertensive treatment, BMI and fasting levels of HDL and LDL cholesterol, triglycerides and glucose. The correlation between metabolites that were significantly associated with either incident CVD or all-cause mortality and traditional CMD risk factors were performed using partial Spearman’s correlation tests, adjusted for age and sex. Due to skewed distributions, metabolite data was log transformed prior to regression analyses. All statistical analyses were performed using R 3.6.0. Meta-analyses were performed in R package meta [32] and partial Spearman’s correlations using R package ppcor [33].

## 5. Conclusions

This is the first study exploring the relationship between a broad range of metabolites and CVD morbidity and mortality in individuals with diabetes. We identify strong associations between 10 plasma metabolites and all-cause mortality highlighting measurable derangement of both central metabolic pathways and nutritional intake several years before pre-mature mortality occurs in patients with diabetes. Our findings should stimulate further studies examining whether these metabolites’ relationship with premature mortality is causal, and thus can contribute to novel therapies prolonging life in diabetes.

## Figures and Tables

**Figure 1 metabolites-10-00315-f001:**
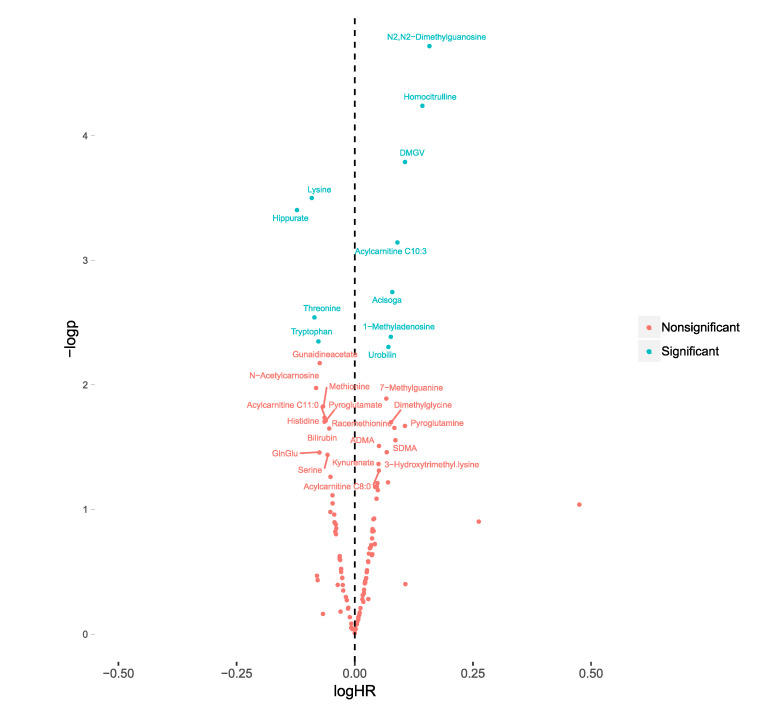
Associations between plasma levels of 112 metabolites and all-cause mortality in participants with diabetes from the Malmö Diet and Cancer-Cardiovascular Cohort (MDC-CC) (*n* = 374) and the Malmö Preventive project (MPP) (*n* = 369). The 10log hazard ratios (logHR) were calculated separately in MDC-CC and MPP from cox regression models (adjusted for age and sex) and subsequently meta-analyzed using inverse variance-weighted meta-analysis, using fixed estimates. LogHR of all-cause mortality is expressed per standard deviation increment of each metabolite. Associations were considered significant at a false-discovery rate <0.05. −logp= −10log of the *p*-value.

**Figure 2 metabolites-10-00315-f002:**
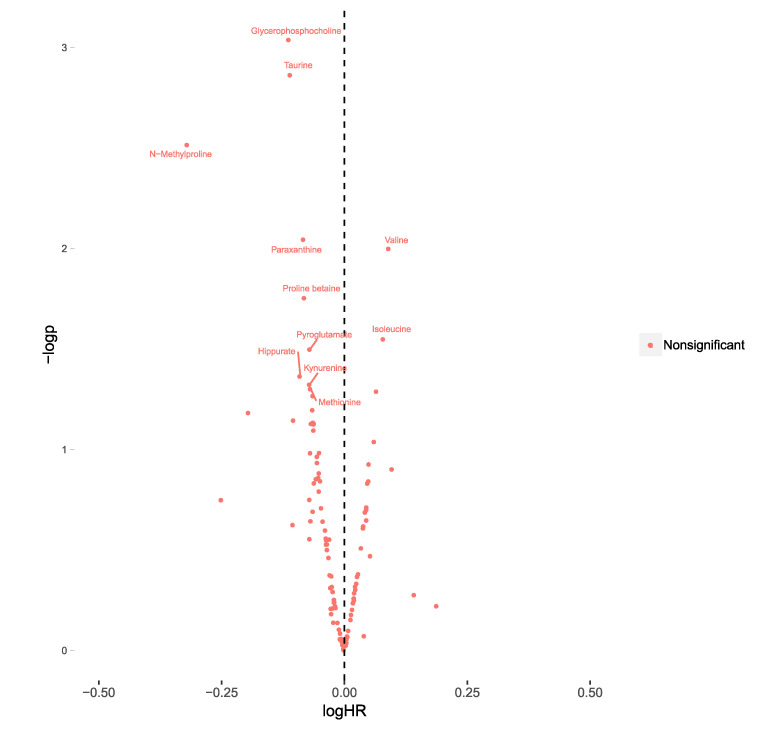
Associations between plasma levels of 112 metabolites and incident cardiovascular disease (CVD) in participants with diabetes from the Malmö Diet and Cancer-Cardiovascular Cohort (MDC-CC) (*n* = 357) and the Malmö Preventive project (MPP) (*n* = 315). The 10log hazard ratios (logHR) were calculated separately in MDC-CC and MPP from cox regression models (adjusted for age and sex) and subsequently meta-analyzed using inverse variance-weighted meta-analysis, using fixed estimates. LogHR of all-cause mortality is expressed per standard deviation increment of each metabolite. Associations were considered significant at a false-discovery rate <0.05. −logp= −10log of the *p*-value.

**Figure 3 metabolites-10-00315-f003:**
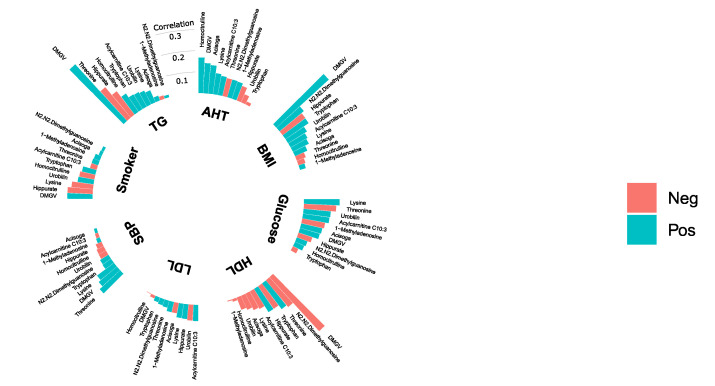
Correlations between plasma levels of 11 plasma metabolites and BMI, smoking status (Smoker), systolic blood pressure (SBP), anti-hypertensive treatment (AHT) and fasting levels of high-density lipoprotein cholesterol (HDL), low-density lipoprotein cholesterol (LDL), triglycerides (TG) and glucose in the Malmö Diet and Cancer-Cardiovascular Cohort (*n* = 374). Correlation coefficients are either partial Spearman’s correlation coefficients (BMI, glucose, HDL, LDL, TG, SBP) or partial Pearson’s correlation coefficients (Smoker and AHT), adjusted for age and sex.

**Figure 4 metabolites-10-00315-f004:**
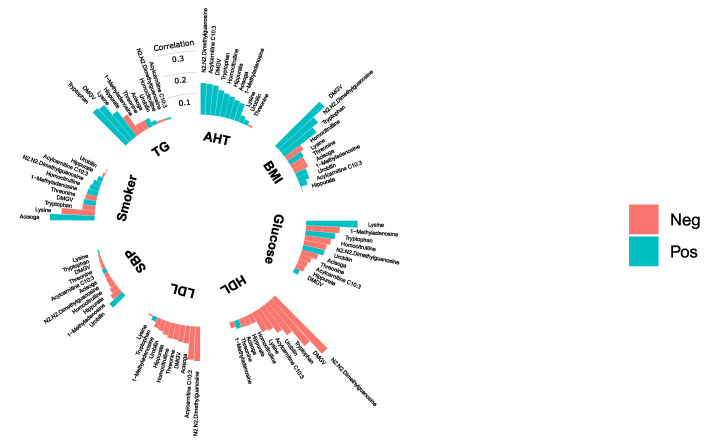
Correlations between plasma levels of 11 plasma metabolites and BMI, smoking status (Smoker), systolic blood pressure (SBP), anti-hypertensive treatment (AHT) and fasting levels of HDL cholesterol (HDL), LDL cholesterol (LDL), triglycerides (TG) and glucose in the Malmö Preventive Project (*n* = 369). Correlation coefficients are either partial Spearman’s correlation coefficients (BMI, glucose, HDL, LDL, TG, SBP) or partial Pearson’s correlation coefficients (Smoker and AHT), adjusted for age and sex.

**Table 1 metabolites-10-00315-t001:** Characteristics of the participants in the Malmö Diet and Cancer-Cardiovascular Cohort (MDC-CC) and Malmö Preventive Project (MPP).

Trait	MDC-CC (*n* = 374)	MPP (*n* = 336)
**Age (years)**	59.5 (±5.6)	70.5 (±5.9)
**Sex (% female)**	44.9	23.6
**Body mass index (BMI) (kg/m^2^)**	28.3 (±4.5)	28.8 (±4.4)
**Fasting glucose (mmol/L)**	7.9 (±3.1)	7.7 (±2.2)
**LDL cholesterol (mmol/L)**	4.2 (±1.0)	3.4 (±1.0)
**HDL cholesterol (mmol/L)**	1.2 (±0.3)	1.3 (±0.4)
**Triglycerides (mmol/L)**	1.8 (±0.8)	1.5 (±0.8)
**Current Smokers (%)**	23.5	19.0
**Prevalent Cardiovascular Disease (%)**	4.5	14.7

Numbers in parenthesis indicate the standard deviations.

**Table 2 metabolites-10-00315-t002:** Associations between plasma metabolites and all-cause mortality in participants with diabetes from the Malmö Diet and Cancer-Cardiovascular Cohort (MDC-CC) and the Malmö Preventive Project (MPP).

Metabolites	HR	*p*
**N2,N2-dimethylguanosine**	1.45 (1.22–1.72)	1.9 × 10^−5^
**DMGV**	1.31 (1.14–1.51)	2.0 × 10^−4^
**Lysine**	0.80 (0.71–0.90)	2.0 × 10^−4^
**Homoctirulline**	1.34 (1.13–1.58)	7.6 × 10^−4^
**Hippurate**	0.78 (0.66–0.92)	3.4 × 10^−3^
**1-methyladenosine**	1.18 (1.04–1.35)	9.4 × 10^−3^
**Acylcarnitine C10:3**	1.19 (1.04–1.36)	0.012
**Tryptophan**	0.85 (0.75–0.97)	0.013
**Urobilin**	1.15 (1.02–1.30)	0.023
**Threonine**	0.86 (0.75–0.99)	0.039
**Acisoga**	1.11 (0.98–1.25)	0.11

Associations between plasma levels metabolites and all-cause mortality in participants with diabetes from the Malmö Diet and Cancer-Cardiovascular Cohort (MDC-CC) (*n* = 374) and the Malmö Preventive project (MPP) (*n* = 369). The hazard ratios (HR) were calculated separately in MDC-CC and MPP from cox regression models (adjusted for age, sex, BMI, Systolic blood pressure, antihypertensive treatment, smoking status and fasting levels of glucose, HDL cholesterol, LDL cholesterol and triglycerides) and subsequently meta-analyzed using inverse variance-weighted meta-analysis, using fixed estimates. The HR of all-cause mortality is expressed per standard deviation increment of each metabolite. Numbers in parenthesis indicate 95% confidence intervals.

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
