# Peer review of "Plasma Metabolites Associate with All-Cause Mortality in Individuals with Type 2 Diabetes"

_metabolites, 2020, doi:10.3390/metabo10080315_

Round 1

Reviewer 1 Report

GENERAL COMMENTS:  The authors outline a prospective metabolomics study to identify predictive plasma biomarkers associated with incident CVD and all-cause morality among high risk diabetic participants. Overall, the manuscript is well written with appropriate use of statistical analyses and data interpretation. However, analytical details of the LC-MS protocols, quality control measures and metabolite authentication and importantly identification need far more attention. As a result, the manuscript in its current version require major revisions to improve overall data transparency.

SPECIFIC COMMENTS:

  1. Methods/Samples: The authors are recommended to include a consort flow diagram (as supplemental or figure in main manuscript) outlining inclusion/exclusion criteria applied for participant selection in the two cohorts given their quite different demographic composition. Clinical outcome measures should also be specified, including standard risk assessment tools (e.g., blood lipid panel) for predicting incident CVD.
  2. Methods/LC-MS: The authors cite an earlier paper (ref. 17) used for amino acid analysis that forms the basis of their plasma metabolomics study. There are two few experimental details provided here for replication. Were all plasma extracts analyzed under positive and negative ion modes? What was a modest number of total metabolites (112) measured by this method that seems to focus primarily on polar./ionic metabolites?
  3. Methods/LC-MS: Similarly, the authors are strongly recommended to follow reporting standards in metabolomics with all metabolites annotated by their accurate mass, retention time, as well as most likely molecular formula and MS/MS spectra notably for clinical significant metabolites identified in this work. How were the proposed structures for all novel metabolites confirmed (level of confidence in identification) especially if standards were lacking. All metabolites should have HMDB identifier or other descriptors given differences in their nomenclature (e.g., acisoga, DMGV) and potential isomeric candidates (dimethylguanosine) as shown in Table 2. A supporting information file should be included in the manuscript that lists all fully annotated 112 plasma metabolites, including their technical precision and frequency of detection in all samples (missing value inputs). This is a major weakness of the present manuscript.
  4. Results/Discussion: All significant metabolites correlated with incident CVD or all cause mortality from Cox regression model should have technical precision information from intermittent analysis of QCs (CV < 20-30%) to demonstrate that they are likely reliably measurable compounds. The authors do mention that such quality control measures were implemented, but no evidence is provided in the results.
  5. Figures: Overall quality of figures needs improvement as the resolution of text is poor.
  6. Results/Discussion: Can the authors consider a panel of plasma metabolites that can further improve the predictive accuracy instead of single metabolites alone? Is there a combination of metabolites or metabolite ratios that have better clinical utility as predictive biomarkers? Also, please provide a comparison with standard risk assessment tools used for predicting incident CVD or all-cause mortality (e.g, fasting glucose, insulin/HOMA, HDL etc.). Would the addition of one or more plasma metabolites further improve upon these classical predictive biomarkers?
  7. Results/Discussion: I presume that the authors were not able to explore potential correlations between circulating metabolites and self-reported dietary patterns from the two cohorts in their work given the important role of diet in type 2 diabetes and CVD? This can be discussed as a study limitation if not available, as well as limited metabolome coverage using their LC-MS platform (notably for lipids).

Author Response

Please see the attatchment

Reviewer 2 Report

The authors presented a very interesting topic of metabolome associated mortality in type 2 diabetic patients. They used LCMS based metabolomics technology analyzed 743 T2DM patients in 2 cohorts and followed up with the CVD new cases and death. They found 11 metabolites significantly associated with all-cause mortality. The research topic is of great interest to the audience and very important to the field. There are some comments on the following:

  1. The important findings should be validated in larger cohorts in future studies.
  2. The authors used two different anticoagulants for these two cohorts (EDTA and citrate). Is there any difference between the metabolome profiles?
  3. The samples were stored for a very long time and there is about a 10-year difference for samples from these two cohorts. Did the authors observe any metabolites degradation over time?
